# Study on the Prevalence of *Pneumocystis jirovecii* as a Causative Agent of Lung Pathology in People with Different Immune Status

**DOI:** 10.3390/biomedicines11071851

**Published:** 2023-06-28

**Authors:** Rumen Harizanov, Nina Tsvetkova, Aleksandra Ivanova, Raina Enikova, Mihaela Videnova, Iskra Rainova, Eleonora Kaneva, Iskren Kaftandjiev, Dimitar Strashimirov, Nina Yancheva-Petrova, Ivan Simeonovski, Viktoria Levterova, Nikolay Yanev

**Affiliations:** 1Department of Parasitology and Tropical Medicine, National Centre of Infectious and Parasitic Diseases, 26 Yanko Sakazov Blvd., 1504 Sofia, Bulgaria; tsvetkova@ncipd.org (N.T.); aleksandra.ivanova@ncipd.org (A.I.); rainaborisova@ncipd.org (R.E.); mvidenova@ncipd.org (M.V.); rainova@ncipd.org (I.R.); kaneva@ncipd.org (E.K.); kaftandjiev@ncipd.org (I.K.); 2Department for AIDS, Specialized Hospital for Active Treatment of Infectious and Parasitic Diseases, Ivan Geshev Blvd. 17, 1431 Sofia, Bulgaria; dstrashimitov@live.com (D.S.); dr.yahcheva@abv.bg (N.Y.-P.); 3Department of Microbiology, National Centre of Infectious and Parasitic Diseases, 26 Yanko Sakazov Blvd., 1504 Sofia, Bulgaria; ivanos@dir.bg (I.S.); vikis@abv.bg (V.L.); 4Department of Bronchology, University Multi-Profile Hospital (UMBAL) for Active Treatment of Lung Diseases “Sveti Ivan Rilski” EAD, Ivan Geshev Blvd. 19, 1431 Sofia, Bulgaria; dr.nikolay.yanev@gmail.com

**Keywords:** *Pneumocystis* pneumonia, immunity, genetic techniques, microscopy

## Abstract

Background: *Pneumocystis* pneumonia (PCP) commonly affects immunocompromised individuals, whereas in immunocompetent persons, it occurs relatively rarely, and in most cases, the *Pneumocystis* infection is detected as an asymptomatic colonization. The present study aimed to establish the prevalence of *Pneumocystis jirovecii* infection in human hosts with different immune status (immunocompromised and immunocompetent), using molecular diagnostic methods, and to compare their diagnostic value with that of classical staining methods. Methods: We used the collected-to-this-moment data from a prospective study on the prevalence of pneumocystosis among the Bulgarian population. Clinical specimens (including throat secretions, induced sputum, tracheal aspirates, and bronchoalveolar lavage) collected from 220 patients suspected of PCP (153 immunocompetent and 67 immunocompromised patients) were examined with microscopic staining methods and real-time PCR for detection of *P. jirovecii.* Results: DNA of the pathogen was detected in 38 (17%) specimens (32 immunocompromised patients and 6 immunocompetent subjects). From all 220 clinical samples examined by staining methods, only five (2%) *P. jirovecii* cysts were detected by the Gomori stain. All patients with PCP were treated with trimethoprim-sulfamethoxazole, but in ten of them (HIV-positive patients), the disease had a fatal outcome. Conclusions: This study is the first in Bulgaria including the main available laboratory methods for diagnosis of human pneumocystosis. Regarding the etiological diagnosis of PCP, in our study the sensitivity of real-time PCR was higher compared to the staining methods. The choice of a method for sample collection and examination has an important role in the efficiency of the laboratory diagnostics.

## 1. Introduction

The first reports of the involvement of *Pneumocystis* in human pathology date back to the 1940s as a pathogen causing pneumonia in malnourished or premature infants [1,2]. In 1942, Van der Meer and Brug described cases of interstitial pneumonia caused by *Pneumocystis* in two infants and one young adult, which ended fatally [3,4]. In 1951, Vanek and Jirovec reported sporadic outbreaks in Europe caused by *Pneumocystis* in premature infants with interstitial plasma cell pneumonia [5]. Until the 1980s, *Pneumocystis* pneumonia (PCP) in the elderly was considered a rare but fatal infection mainly among patients with acute leukemia and other hematological malignancies [6]. In the 1980s, after the onset of the global epidemic of the human immunodeficiency virus (HIV) infection, interest in PCP increased dramatically as it was one of the leading co-infections and the cause of death in patients with acquired immune deficiency syndrome (AIDS). Due to advances in access to antiretroviral therapy (ART) and routine prophylaxis against PCP, its incidence in the HIV-infected population has decreased in most industrialized countries. However, PCP still remains the most common opportunistic infection among AIDS patients in many countries [7,8,9].

Several studies have reported the increasing number of PCP cases in non-HIV-infected patients due to the growing number of people diagnosed with an underlying condition and receiving immunosuppressive therapy with corticosteroids and cytotoxic agents [10,11]. At high risk for *P. jirovecii* infection were patients with malignant diseases, conditions after transplantation of stem cells or of solid organs, and those taking immunosuppressive medications for autoimmune disorders [12,13,14,15,16,17,18,19]. The most commonly reported underlying disorder (for the development of PCP) in the study conducted by Liu et al. was hematological malignancies (29.1%). Other disorders included autoimmune disease (20.1%) and transplant recipients (e.g., organ or bone marrow transplantation (14.0%), and solid tumors (6.0%) [20].

*Pneumocystis* may be present in the respiratory system without leading to the clinical presentation of severe pneumonia. Detection of *Pneumocystis* in individuals without clinical symptoms is defined as colonization and is increasingly recognized as a serious, worldwide public health concern [21]. Pulmonary colonization with *P. jirovecii* is common in immunocompromised patients but is less frequent among immunocompetent individuals with lung disease [22,23,24]. A relationship between *P. jirovecii* colonization and development of pulmonary diseases in non-HIV-infected patients has been reported in the literature. Patients with interstitial lung diseases were more frequently diagnosed with *P. jirovecii* colonization compared to those with chronic obstructive pulmonary disease (COPD) and patients with acute exacerbations of COPD (85% vs. 67 and 43.3%, *p*  <  0.01) [25]. Association between older age and higher prevalence of *Pneumocystis* colonization has been assumed. Previous studies indicate that the elderly patients with different kinds of pulmonary disorders play an important role in transmission of the infection with this fungus to high-risk individuals [26].

In Bulgaria, pneumocystosis has been the subject of scientific interest for years. *P. jirovecii* was found in young children with pneumonia and in patients with AIDS [27,28,29]. The first case of pneumocystosis associated with AIDS in our country was diagnosed at the National Centre of Infectious and Parasitic Diseases [30,31,32]. In a study (covering the period 1986–2000) of the prevalence and features of different types of opportunistic infections in HIV-infected and AIDS patients in Bulgaria, *Pneumocystis* was detected in 26 (16.25%) of 160 patients by direct microscopic examination (staining methods and immunofluorescence assays). The clinical manifestation of pneumocystosis was represented by pneumonia with a disseminated type of infection in one case [33]. In another study, conducted between 1993 and 2003, pneumocystosis was diagnosed in 6.06% of the examined (*n* = 165) patients [34]. A study by Yancheva et al. (2020) revealed high PCP mortality (46.3%) in HIV-infected patients with severe immune suppression, despite the etiological therapy administered [35].

The present study aimed to establish the prevalence of pneumocystosis in different groups of immunocompromised and immunocompetent individuals using molecular diagnostic methods and to compare their diagnostic value with that of classical staining methods.

## 2. Materials and Methods

### 2.1. Study Design

This article is based on data from a prospective study on the prevalence of pneumocystosis among the Bulgarian population, beginning in January 2019.

### 2.2. Ethical Considerations

The study was reviewed and approved by the institutional review board (IRB) 00006384 and informed consent was obtained from the patients. No information that could reveal the identity of the patients who participated in the study was used.

### 2.3. Patients and Samples

The study included admitted patients from various hospitals in the country suspected of PCP. Patients were selected by their treating physicians based on clinical and imaging data. Our role was expressed in the examination of the clinical samples by applying the diagnostic techniques implemented in the Department of Parasitology and Tropical Medicine at the National Centre of Infectious and Parasitic Diseases and subsequent consultations regarding the therapeutic behavior. For this reason, there may be some limitations in the exact selection of patients.

Clinical specimens (including throat secretion, induced sputum, tracheal aspirate, and bronchoalveolar lavage (BAL)) collected from 220 patients suspected of having pneumocystosis (Group 1—immunocompetent patients and Group 2—patients with compromised immune system) were examined.

Group 1 included a total of 153 individuals presenting with cough (*n* = 106), evidence of unspecified pneumonia (*n* = 23), shortness of breath (*n* = 4), respiratory failure (*n* = 3), hemoptysis (*n* = 3), bronchitis (*n* = 2), fatigue (*n* = 2), respiratory distress syndrome (*n* = 1), pharyngitis (*n* = 1), lung abscess (*n* = 1), and 7 with coronavirus disease (COVID-19)-related pneumonia.

Group 2 consisted of 67 individuals—47 with HIV infection and 20 on immunosuppressive therapy (including hematological disease, *n* = 7; interstitial pulmonary fibrosis, *n* = 3; nephrotic syndrome, *n* = 3; bronchiectasis, *n* = 3; solid organ transplantation, *n* = 2; asthma, *n* = 1; and disseminated lupus, *n* = 1).

### 2.4. Methods for Detection of Pathogen

#### 2.4.1. Real-Time Polymerase Chain Reaction (PCR) for Qualitative and Quantitative Detection of *P. jirovecii*

For extraction and purification of *P. jirovecii* Deoxyribonucleic acid (DNA) from clinical samples, PureLink™ Genomic DNA Mini Kit (Life Technologies Corporation, Carlsbad, CA 92008 USA) was used. Amplification of the gene encoding the mitochondrial large subunit of ribosomal ribonucleic acid (mtL SU rRNA) was performed by using RIDA^®^GENE kit (r-biofarm AG, Pfungstadt, Germany) according to the manufacturer’s instructions.

#### 2.4.2. Staining Methods for Detection of *P. jirovecii*

Three staining methods for direct detection of *P. jirovecii* were applied. Six smears from each clinical material (throat secretion, induced sputum, tracheal aspirate, or bronchoalveolar lavage) of the PCP-suspected patient using clean glass microscope slides were prepared. After drying, two smears were stained by each of the three methods.
Romanowski–Giemza staining (for trophozoites and cysts of *P. jirovecii).* Commercial Giemsa stain, modified solution (Sigma-Aldrich, St. Louis, MO, USA), was used. Dried thin smears were fixed with methyl alcohol for 5–10 min, dried, stained with a working solution of Giemsa stain for 20–22 min (the exposure was determined during the initial testing of the stain), washed with tap water, and allowed to dry in a vertical position at room temperature.Toluidine blue staining (selective method for cysts of *P. jirovecii*). The thin smears from each clinical material were immersed for 5 min in sulfate reagent (prepared by mixing 25 mL diethyl ether and 25 mL concentrated sulfuric acid), rinsed with tap water, and stained with toluidine blue solution for 3 min. Differentiation was then performed in 2 shifts of isopropyl alcohol for 15–30 s, lightening with xylene, and finally, drying.Staining with methenamine-silver nitrate according to Gomori (for cysts of *P. jirovecii*). The method is considered the “gold standard” for microscopic visualization of *P. jirovecii* cysts. Microscopy Methenamine silver plating kit acc. to Gomori (Cat. No. 1.00820.0001; Merck KGaA, 64271 Darmstadt, Germany) was used. The dried smears of the relevant clinical material were fixed for 30 min in 3.5% formalin and stained according to the manufacturer’s protocol. The color of the cyst wall varies from gray to black (their surface membranes are visible).The samples were examined under a light microscope (Euromex IS.1153-Pli, Papenkamp 20, 6836 BD Arnhem, The Netherlands) at 400× and 1000× magnification and visualized using color digital camera (Euromex DC.6000s, Papenkamp 20, 6836 BD Arnhem, The Netherlands).

### 2.5. Statistical Analysis

To compare the sensitivity of our pathogen detection methods, we used GraphPad Prism 9 statistical software (GraphPad Software Inc.). Fisher’s exact test was used for clinical data analysis, with *p* values lower than 0.05 accepted as statistically significant. Because our data provide a basis for evaluating diagnostic tests, we determined metrics such as sensitivity and specificity using the Wilson/Brown hybrid method.

## 3. Results

We applied real-time PCR targeting the mtL SU rRNA gene of *P. jirovecii* for qualitative and quantitative detection of the pathogen from clinical specimens of the patients included in the study. *P. jirovecii* DNA was detected in the specimens of 38 (17%) out of 220 subjects examined. Positive PCR results were obtained in specimens of 6 patients with pneumonia of the 153 individuals of Group 1. In Group 2, amplification of a fragment of the target gene was obtained in 26 of the HIV-infected patients and 6 patients receiving suppressive therapy (Table 1).

Commercial kit RIDA^®^GENE *Pneumocystis jirovecii* (r-biofarm AG, Germany), containing standards with a certain number of copies of *P. jirovecii* (Standard A: 101 copies/µL, Standard B: 103 copies/µL, Standard B: 105 copies/µL; analytical sensitivity: ≥10 DNA copies per reaction) was used for pathogen-load assessment in the tested samples. In the post-treatment period for *Pneumocystis* pneumonia, four patients of the immunocompromised group and three patients without immunosuppression were additionally tested in follow-up studies for assessment of their response to the therapy. The reason for these additional tests was the patient’s continuing complaints of shortness of breath, cough, and ongoing fever. Except for one HIV-infected patient, in all others, we found that the control sample taken within one month after the end of therapy did not contain *P. jirovecii* DNA. The complaints of the patient with HIV infection, whose control sample had a positive PCR result one month after the examination of the primary clinical specimen (induced sputum), continued despite the prescribed etiological treatment. On the background of assigned secondary prophylaxis with trimethoprim/sulfamethoxazole (TMP/SMX), a total of nine control tests a month apart from each other were performed for monitoring the response of the treatment, and only in the last sample no DNA of the pathogen was detected [36].

All 220 clinical samples of patients suspected of having pneumocystosis from Groups 1 and 2 were examined by staining methods. In patients from Group 1, staining methods showed no evidence of *P. jirovecii* cysts, while in five of Group 2 patients, the presence of *P. jirovecii* cysts was detected only by the Gomori’s methenamine-silver stain (Table 1, Figure 1).

The results of Gomori’s staining, quantification of DNA load, and PCR cycle (Ct value), in which a fluorescent signal was reported as a result of multiplication of the target region of DNA molecule in the test samples are presented in Table 2.

In some patients, we also performed a comparative study on the concentration of *P. jirovecii* DNA depending on the type of clinical specimen (Table 3).

## 4. Discussion

Cases of HIV-associated PCP are reported at fluctuating rates throughout the world. While the clinical manifestation of PCP in HIV-positive patients is well known and consists most often of the triad of dyspnea, fever, and cough, the presentation of PCP in HIV-negative patients is atypical and occurs suddenly with oxygen desaturation and rapid death if left untreated [37]. This pathology is a serious public health problem not only because of the severity of the disease but also because PCP is a life-threatening condition in HIV-negative immunocompromised patients [38]. Another public health problem is the number of colonized patients in hospital wards, where other patients may be at high risk of infection or colonization [39].

According to the literature, the diagnosis of PCP or colonization depends on a complex algorithm based on the patient’s medical history, laboratory and radiological data, treatment, and clinical evolution of the patient’s condition. If a positive microscopic examination leads to a high probability of PCP, a negative quantitative PCR (qPCR) result cannot rule out the diagnosis, especially in HIV-negative patients. In most cases, qPCR is sensitive enough to allow the diagnosis of PCP in HIV patients; however, the presence of a gray area of Ct values prevents this analysis from becoming a reference method [40]. In the case of Ct values in the gray area, the physician will have to choose between prophylactic or active treatment according to the clinical parameters of the disease and the patient’s condition [38].

With this study, we demonstrate that the use of real-time PCR can significantly improve the differential diagnosis in patients suspected of having *P. jirovecii* infection and clarify the infection epidemiology in immunocompromised patients and those without immunosuppression.

Our data show that of the 153 examined individuals without evidence of compromised immunity, 4% (*n* = 6) tested positive for *P. jirovecii* DNA. All of them were with severe pneumonia and PCP-specific radiological findings. While in 66.7% (*n* = 4) of them the condition can be explained by their infant (0–12 months) and child age (14 years), the remaining 33.3% (*n* = 2) were adults over 18 years old, and the development of PCP was difficult to explain. The most significant risk factors for PCP in HIV-free patients are the use of glucocorticoids and the presence of cell-mediated immune defects, which lead to changes in lung surfactant, thus predisposing the patient to pneumonia [41]. Previous studies have documented PCP in immunocompetent individuals [42,43,44,45,46]. A case of PCP in a non-HIV-infected Indian patient who has not undergone previous glucocorticoid treatment was described by Koshy et al. (2015) [42]. Kawame et al. (2022) reported a case of PCP in an immunocompetent individual without evidence for local or systemic immunodeficiency who had a subacute onset and bilateral central consolidation shown by the chest radiograph [43]. There have been several reports of PCP in patients without underlying immunosuppressive disease. A study by Kano et al. described five patients who developed PCP without any underlying immunosuppressive conditions, and in their literature review, they identified only 11 other reported cases [44]. However, the exact mechanisms that lead to the development of PCP in patients without evidence of immunosuppressive status remain unclear [45].

From individuals included in the study with compromised immunity and lung pathology, 47.8% (*n* = 32) showed the presence of *P. jirovecii* DNA. Of them, 26 (38.8%) were HIV infected and 6 (8.96%) had other immunosuppressive conditions (Table 1). The age distribution shows a prevalence of patients over 18 years of age (*n* = 29, 90.6%), while three of the patients (9.4%) were in the age group from 1 to 9 years. One of the children was HIV-positive, and two were non-HIV-infected. One child developed PCP symptoms on the background of oncohematological disease, and the other was on long-term corticosteroid therapy for nephrotic syndrome. In the group of immunocompromised patients, the distribution by sex showed a predominance of males (*n* = 26, 81.3%) compared to females (*n* = 6, 18.7%). In general, our data are similar to those in the literature for people at risk for developing PCP pneumonia [46].

Regarding the etiological diagnosis of *Pneumocystis* pneumonia, staining with methenamine-silver nitrate according to Gomori is considered as the “gold standard” for microscopic visualization of *P. jirovecii* cysts. However, our study gives us reason to consider that the real-time PCR is more useful for diagnostic purposes than the staining methods (Table 1). The statistical analysis of the data showed 88.37% sensitivity of the PCR test, while for the microscopy of stained preparations, this indicator was 11.63%. In terms of specificity, these indicators are respectively 54.16% and 45.84%. Patients with compromised immunity were more likely to have detectable cysts of the pathogen in obtained from them clinical specimens than those without immunosuppression. The data from our study prove the existence of a correlation between the levels of the pathogen load and the detectability of *P. jirovecii* cysts with staining methods (Table 2).

The choice of a method for sample collection and examination has an important role in the efficiency of the laboratory diagnostics. Results showed that various clinical specimens (induced sputum, tracheal aspirate, and BAL) can be tested to diagnose pneumocystosis. Our initial observations revealed that in infants and young children, the tracheal aspirate is a good enough clinical sample for genetic testing.

In Bulgaria, the first-line agent for the treatment of PCP is trimethoprim-sulfamethoxazole, regardless of the patient’s immune status, while the use of additional drugs and oxygen therapy depends on the patient’s condition and is in accordance with generally accepted international guidelines.

In the post-treatment period for *Pneumocystis* pneumonia, four patients in the immunocompromised group and three patients without immunosuppression had follow-up studies to assess their response to the therapy. The reason for this was continuing complaints such as shortness of breath, cough, and fever. With the exception of one HIV-infected patient, in all others, we found that the control sample taken within one month after the end of therapy did not contain *P. jirovecii* DNA.

Unfortunately, in ten of the patients (HIV-positive people) the disease was fatal. The mortality rate among people with PCP for the period studied by us was 26.3%, and our data are similar to the literature [47,48,49].

## 5. Conclusions

This study is the first in the country including the main available laboratory methods for the diagnosis of human pneumocystosis in Bulgaria. Regarding the etiological diagnosis of PCP, based on our study, the sensitivity of real-time PCR is higher compared to the staining methods. Patients with compromised immunity were more likely to have detectable cysts of the pathogen in obtained from them clinical specimens than those without immunosuppression. Results showed that various clinical specimens (induced sputum, tracheal aspirate, and BAL) are suitable for diagnosis of *Pneumocystis* infection. Our initial observations revealed that in infants and young children, tracheal aspirate is a good enough clinical sample for genetic testing. The choice of a method for sample collection and examination plays an important role in the efficiency of the laboratory diagnostics.

## Figures and Tables

**Figure 1 biomedicines-11-01851-f001:**
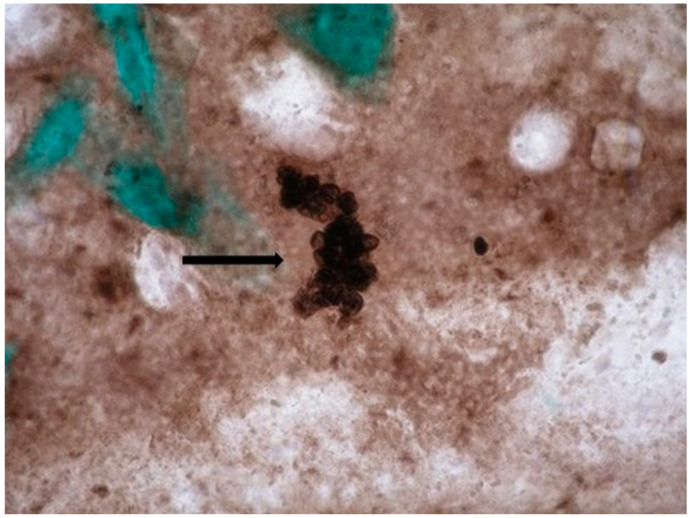
Sputum smear of HIV+ patient, stained with methenamine-silver nitrate (according to Gomori). A cluster of black-stained oval and rounded cysts of *P. jirovecii* (arrow). Light microscopy, magnification 1000×.

**Table 1 biomedicines-11-01851-t001:** Demographic, immunological, and clinical data for study participants.

Demographic Data	Age Groups (Range)				Gender		Total N (%)
	0–12 Months	1–9 Years	10–18 Years	>18 Years	Male Sex	Female Sex	
No of cases	25	36	31	128	137	83	220
Real-time PCR positive	3	2	2	31	32	6	38 (17.3%)
Real-time PCR negative	22	34	29	97	105	77	182 (82.7%)
Light microscopy (RG ^1^/TB ^2^/GMS ^3^), positive	0	0	0	5	5	0	5 (2.3%)
Light microscopy, negative	25	36	31	123	132	83	215 (97.7%)
Groups distributed by immunological status and clinical presentation (primary diagnosis)							
Group 1—patients without data of immunosuppression	23	34	26	70	86	67	153
pneumonia	7	1	1	14	16	7	23 (15%)
respiratory distress syndrome	0	0	1	0	1	0	1 (0.7%)
pharyngitis	0	0	0	1	1	0	1 (0.7%)
respiratory failure	1	0	0	2	1	2	3 (1.9%)
dyspnea	0	0	1	3	1	3	4 (2.6%)
pulmonary abscess	0	0	0	1	1	0	1 (0.7%)
bronchitis	0	1	0	1	0	0	2 (1.3%)
fatigue	0	0	0	2	2	0	2 (1.3%)
hemoptysis	0	0	0	3	1	2	3 (1.9%)
cough	15	32	23	36	56	50	106 (69.3%)
COVID-19	0	0	0	7	4	3	7 (4.6%)
Real-time PCR positive	3	0	1	2	6	0	6 (3.9%)
Real-time PCR negative	20	34	25	68	80	67	147 (96.1%)
Light microscopy, positive	0	0	0	0	0	0	0
Group 2—patients with Compromised immune system	2	2	5	58	51	16	67
HIV infection	0	1	0	46	42	5	47 (70%)
hematological malignancy	1	0	1	5	4	3	7 (10.5%)
interstitial pulmonary fibrosis	0	0	0	3	2	1	3 (4.5%)
nephrotic syndrome	0	1	2	0	1	2	3 (4.5%)
solid organ transplantation	1	0	1	0	2	0	2 (3%)
long-term use of inhaled corticosteroids due to bronchiectasis and asthma	0	0	0	4	0	4	4 (6%)
disseminated lupus	0	0	1	0	0	1	1 (1.5%)
Real-time PCR positive	0	2	1	29	26	6	32 (47.8%)
Real-time PCR negative	2	0	4	29	25	10	35 (52.2%)
Light microscopy, positive	0	0	0	5 (GMS ^3^)	5	0	5 (7.5%)
Light microscopy, negative	2	2	5	53	46	16	62 (92.5%)

^1^ RG—Romanowski–Gimza staining; ^2^ TC—toluidine blue staining; ^3^ GMS—Gomori’s methenamine-silver stain.

**Table 2 biomedicines-11-01851-t002:** Comparison of data from studies with the Gomori’s methenamine-silver stain and the real-time PCR (the specific target gene of *Pneumocystis jirovecii* is mtL SU rRNA) of 5 patients with positive results by both techniques.

Patients	Staining Method Specimen Type-Induced Sputum	Real-Time Quantitative PCR		Ct
	GMS	*P. jirovecii* DNA Concentration (Copies/µL)		
		In 1 µL of the Reaction Solution	In 200 µL of the Initial Sample	
P1 HIV+	Clusters of cysts	5.035 × 10^5^	1.007 × 10^8^	18.074
P2 HIV+	Clusters of cysts	4.669 × 10^5^	9.338 × 10^7^	18.176
P3 HIV+	Single cysts	2.179 × 10^1^	4.358 × 10^3^	31.566
P4 HIV+	Single cysts	5.790 × 10^1^	1.158 × 10^4^	30.254
P5 HIV+	Single cysts	4.703 × 10^2^	9.406 × 10^4^	27.441

**Table 3 biomedicines-11-01851-t003:** Concentration of *P. jirovecii* DNA depends on the type of clinical specimen.

Type of Clinical Specimen	Patients/Age Group	Real-Time Quantitative PCR		Ct
		Concentration of *P. jirovecii* DNA (Copies/µL)		
		In 1 µL of the Reaction Solution	In 200 µL of the Initial Sample	
Tracheal aspirate	A 4-month-old baby with pneumonia	0.8123 × 10^3^	0.162480 × 10^6^	35.37
	A 6-month-old baby with severe interstitial pneumonia	359.6 × 10^3^	71.92 × 10^6^	26.67
Bronchoalveolar lavage	A 60-year-old man with interstitial pulmonary fibrosis	1.265 × 10^3^	253 × 10^6^	24.87
	A 45-year-old man with bilateral interstitial pneumonia	87.52 × 10^3^	17.504 × 10^6^	29.69

## Data Availability

The datasets generated during the current study are available from the corresponding author upon reasonable request.

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
