# Peer review of "Study on the Prevalence of Pneumocystis jirovecii as a Causative Agent of Lung Pathology in People with Different Immune Status"

_biomedicines, 2023, doi:10.3390/biomedicines11071851_

Round 1
Reviewer 1 Report
The title of the manuscript is comprehensive.
English language has good quality.
Not all keywords were selected from the MeSH (Medical Subject Headings) terminology.
Some abbreviations used throughout the text without prior explanation.
The tables and figure meet the required standards.
Please in the section «Introduction» add information about relevant worldwide research on this topic to expand the introduction and justify the significance of the study.
Please in the section «Materials and methods» provide details on the location and method of patient selection, as well as the inclusion and exclusion criteria used in the study.
Please in the section «Results» include information about statistic methods and software, especially which were used to compare the sensitivity of real-time PCR and the staining methods.
Approximately 80% of the references in the bibliography are over 5 years old. It is recommended to update and expand the list of references by including more recent and pertinent sources.
Author Response
Dear Reviewer 1,
On behalf of the author's collective, I would like to thank you for the critical comments and advice that we believe will improve the quality of the text of the article.
“The title of the manuscript is comprehensive.”
“English language has good quality.”
“Not all keywords were selected from the MeSH (Medical Subject Headings) terminology.” – Corrected
“Some abbreviations used throughout the text without prior explanation.” – Corrected
“The tables and figure meet the required standards.”
“Please in the section «Introduction» add information about relevant worldwide research on this topic to expand the introduction and justify the significance of the study.” - In the Introduction section, we have added new texts to emphasize the importance of the issue.
“Please in the section «Materials and methods» provide details on the location and method of patient selection, as well as the inclusion and exclusion criteria used in the study.” – Corrected
“Please in the section «Results» include information about statistic methods and software, especially which were used to compare the sensitivity of real-time PCR and the staining methods.” – Included
„Approximately 80% of the references in the bibliography are over 5 years old. It is recommended to update and expand the list of references by including more recent and pertinent sources.“ - We have included new literature sources, most of them published after 2020.
Reviewer 2 Report
The article submitted for review seems to me to be adequate, providing a series with a large number of patients infected with pneumocystis jirovecii.
The method is appropriate and can provide the reader with information on the diagnosis of the pathology studied.
I would accept the article for publication in the journal in its current version.
English quality is acceptable
Author Response
Dear Reviewer 2,
On behalf of the author's collective, I would like to thank you for your time and kind comment on our article.
Round 2
Reviewer 1 Report
Thanks for considering my comments.